# Domesticated Populations of *Codium tomentosum* Display Lipid Extracts with Lower Seasonal Shifts than Conspecifics from the Wild—Relevance for Biotechnological Applications of this Green Seaweed

**DOI:** 10.3390/md18040188

**Published:** 2020-03-31

**Authors:** Felisa Rey, Paulo Cartaxana, Tânia Melo, Ricardo Calado, Rui Pereira, Helena Abreu, Pedro Domingues, Sónia Cruz, M. Rosário Domingues

**Affiliations:** 1CESAM (Centre for Environmental and Marine Studies), Department of Chemistry, University of Aveiro, Campus Universitário de Santiago, 3810-193 Aveiro, Portugal; taniamelo@ua.pt (T.M.); mrd@ua.pt (M.R.D.); 2Mass Spectrometry Centre & QOPNA & LAQV-REQUIMTE, Department of Chemistry, University of Aveiro, Campus Universitário de Santiago, 3810-193 Aveiro, Portugal; p.domingues@ua.pt; 3ECOMARE, CESAM (Centre for Environmental and Marine Studies), Department of Biology, University of Aveiro, Campus Universitário de Santiago, 3810-193 Aveiro, Portugal; pcartaxana@ua.pt (P.C.); rjcalado@ua.pt (R.C.); sonia.cruz@ua.pt (S.C.); 4ALGAplus- Production and Trading of Seaweed and Derived Products Ltd., Lda., 3830-196 Ílhavo, Portugal; rgpereira@algaplus.pt (R.P.); helena.abreu@algaplus.pt (H.A.)

**Keywords:** antioxidant activity, bioactivity, fatty acids, glycolipids, IMTA, lipidomics, macroalgae, mass spectrometry, phospholipids, photosynthetic pigments

## Abstract

In the last decades, the use of algae in biotechnology and food industries has experienced an exponential growth. *Codium tomentosum* is a green macroalgae with high biotechnological potential, due to its rich lipidome, although few studies have addressed it. This study aimed to investigate the seasonal changes in lipid and pigment profiles of *C. tomentosum*, as well as to screen its antioxidant activity, in order to evaluate its natural plasticity. Samples of *C. tomentosum* were collected in two different seasons, early-autumn (September/October) and spring (May), in the Portuguese coast (wild samples), and in a land-based integrated multitrophic aquaculture (IMTA) system (IMTA samples). Total lipid extracts were analysed by LC–MS, GC–MS, and HPLC, and antioxidant activity was screened through free radical scavenging potential against DPPH and 2,20-azino-bis-3-ethylbenzothiazoline-6-sulfonic acid (ABTS) radicals. Wild samples showed a high seasonal variability, modifying their lipidome and pigment profiles according to environmental shifts, while IMTA samples showed a relatively stable composition due to early-stage culturing in controlled conditions. The lipids that contributed the most to seasonal discrimination were glycolipids (monogalactosyl diacylglycerol - MGDG and digalactosyl diacylglycerol - DGDG) and the lyso forms of phospholipids and glycolipids. Lipid extracts showed antioxidant activity ranging from 61 ± 2 to 115 ± 35 µmol Trolox g^−1^ of lipid extract in DPPH assay and from 532 ± 73 to 927 ± 92 µmol Trolox g^−1^ of lipid extract in ABTS assay, with a more intense antioxidant activity in wild spring samples. This study revealed that wild specimens of *C. tomentosum* presented a higher plasticity to cope with seasonal environmental changes, adjusting their lipid, pigment, and bioactivity profiles, while IMTA samples, cultured under controlled conditions, displayed more stable lipidome and pigment compositions.

## 1. Introduction

Oceans host an impressive wealth of life forms that harbor unique biochemical diversity. Several studies have recently confirmed the high potential of marine organisms as source of bioactive compounds, with interest for the food, fodder, pharmaceutical, biomedical and cosmetic industries [1,2,3,4]. Algae represent some of the most abundant organisms in the marine environment, playing a relevant ecological role on a global scale, as O_2_ producers, CO_2_ sequesters, primary producers, nutrient recycling, and micro-ecosystems. Edible seaweeds are known to be rich in bioactive metabolites, some of them not present in land plants [5]. These metabolites have relevant bioactivities, such as antioxidants [1,2], antimicrobial [4], antigenotoxic [6], and anti-tumorigenic activities [7], holding great potential for biotechnological applications, as food additives [8] and for the production of bioplastics [9], among other high-end uses. Bioactivity has been identified in compounds that are essential for macroalgae metabolism, such as lipids and pigments [10,11,12], but also in products originating from their secondary metabolism [4,13]. Metabolism is strongly influenced by biotic and abiotic factors that promote modifications in biochemical composition, by stimulating or inhibiting the biosynthesis of biologically functional molecules. Algal metabolism is dependent on several factors that influence their metabolome: physiological status, presence/absence of epiphytic organisms, growth habitat, environmental conditions (e.g., temperature, light, and nutrients) or seasonal fluctuations [14,15,16,17]. In order to survive under shifting environments, cells promote metabolic adaptations according to environmental growth conditions and habitats [18]. The understanding of this chemoplasticity is important for the valorization of algae in different applications.

In occidental societies, edible seaweeds do not constitute a significant portion of the human diet, although its consumption has increased in the last years due to the high valorization of oriental cultures and the recognition of the functional benefits of these "plants" from the sea. Progressively, seaweeds are being associated with functional and fortified foods due to the presence of natural bioactive compounds, trace elements, and balanced fatty acid profiles [4,14,19,20,21]. The new challenge of marine chemistry is to understand the natural plasticity of the biochemical composition in marine populations by monitoring them along spatial gradients and variable temporal scales, in order to get the best out of their potential as nutritional food and valuable sources of marine bioactive compounds in a sustainable way.

*Codium tomentosum* (Stackhouse, 1797) is a green marine macroalgae (*Phylum* Chlorophyta) native to the north east Atlantic coast. This macroalga inhabits rock pools and lower seashores and is persistent throughout the year [22,23]. It presents in its composition lipids with nutritional and health benefits [21], and organic acids and volatile compounds with antioxidant properties [13]. Organoleptic properties and composition of the genus *Codium* make these macroalgae appreciated in the gourmet cuisine and cosmetic industry, with the latest using its extracts as a skin protecting agent in commercial formulations [21,24,25]. Lipidomic characterization showed that *C. tomentosum* has a high lipid content (~10%) in comparison to other seaweeds, which usually display between 1% and 2% of lipid content [15,20]. Additionally, this species also showcases some interesting polar lipids that can be putative bioactive phytochemicals, contributing to its valorization [21]. However, seasonal plasticity of the algal lipidome as well as on pigment composition, both relevant groups of metabolites for the valorization of this macroalgae, have not yet been explored.

Thus, the aim of this study is to ascertain how the biochemistry of *C. tomentosum* is shaped by different growth conditions and habitats, analyzing and characterizing its lipid extracts and antioxidant activity. Total lipid extracts, of algae collected from three different sites and in two seasons, were used to identify spatial and temporal changes in lipid, fatty acid and pigment profiles and antioxidant activity. High-resolution hydrophilic interaction liquid chromatography–mass spectrometry and tandem mass spectrometry (HILIC–MS and MS/MS) were used to study the profile of polar lipids, gas chromatography–mass spectrometry (GC–MS) to study the profile of total esterified fatty acids and high performance liquid chromatography (HPLC) to study pigment profiles. Furthermore, antioxidant activity of lipid extracts was screened through the free radical scavenging potential against α,α-diphenyl-β-picrylhydrazyl (DPPH) and 2,20-azino-bis-3-ethylbenzothiazoline-6-sulfonic acid (ABTS) radicals.

## 2. Results

### 2.1. Seasonal Changes in the Polar Lipidome

Total lipid extracts of wild *C. tomentosum* samples presented significantly higher amounts of lipids in autumn (Aguda 57.34 ± 6.86 and Barra 66.15 ± 11.57 µg lipid mg DW^−1^) than in spring (Aguda 38.53 ± 1.64 and Barra 40.16 ± 2.12; µg lipid mg DW^−1^). On the contrary, these seasonal differences were not identified in domesticated samples collected at an integrated multitrophic aquaculture (IMTA) system (autumn 48.33 ± 2.06 and spring 46.77 ± 2.52 µg lipid mg DW^−1^) (Figure 1a).

Lipidomic analysis of total lipid extracts by HILIC–LC MS and MS/MS allowed the identification of 296 molecular ions, belonging to three main groups of polar lipids, namely glycolipids (71 galactolipid and 30 sulfolipid molecular ions), phospholipids (151 molecular ions), and betaine lipids (44 molecular ions) (Appendix A). A complete list of identified lipid classes and species is shown as Appendix A.

Peak areas obtained after integration of extracted ion chromatograms (XIC) of each lipid ion were normalized by the internal standard of each lipid class. The analysis of normalized XIC areas showed different results depending on the sampling site. In Barra, significant increments in most lipid classes were recorded in spring samples, while in the other two sites the differences were not always significant (Figure 2). Regarding glycolipid profiles, the most abundant glycolipid class depended on sample type (wild vs domesticated). In wild samples, monogalactosyl diacylglycerol (MGDG) and sulfoquinovosyl diacylglycerol (SQDG) were the most abundant glycolipid class in autumn and spring, respectively, while IMTA samples presented SQDG as the most abundant in both seasons (Figure 2a). Phospholipidome of wild samples showed phosphatidylglycerol (PG) as the most abundant phospholipid class, while in IMTA samples phosphatidylcholine (PC) and PG showed similar abundances (Figure 2b). In IMTA samples, especially those collected in Autumn showed higher normalized peak areas of most glycolipid and phospholipid classes, than wild samples. Diacylglyceryl 3-O-4´-(N,N,N-trimethyl) homoserine (DGTS) was the betaine lipid class most abundant in all samples (Figure 2c). Considering the lysolipids from the above reported classes, wild specimens presented a significant increment in the normalized peak areas of lysolipids in spring, while in samples originating from IMTA the tendency was inverse, especially in lyso-phospholipids (Figure 2).

Normalized peak areas of all molecular ions identified in the lipidome of *C. tomentosum* were used for the analysis of hierarchical clustering and principal component analyses (PCA). Both analyses showed an association of wild samples according to sampling season, with this being closer for wild spring samples (Figure 3). Nevertheless, hierarchical clustering analysis positioned domesticated samples in the opposite sampling season (i.e. autumn IMTA samples were grouped with spring wild samples and spring IMTA samples with autumn wild samples). The heat map plotted in Figure 4 features the 25 molecular ions that more contributed to the seasonal separation of the samples. The 25 lipid species that mostly contribute to the seasonal discrimination included 8 phospholipids [1 PG, 1 PC, 3 lyso-phosphatidylcholines (LPC), and 3 lyso-phosphatidylethanolamine (LPE)], 15 glycolipids [5 digalactosyl diacylglycerol (DGDG), 6 MGDG, 2 digalactosyl monoacylglycerol (DGMG), and 2 monogalactosyl monoacylglycerol (MGMG)] and 2 betaine lipids [1 DGTS and 1 monoacylglyceryl 3-O-4´-(N,N,N-trimethyl) homoserine (MGTS)]. Overall, glycolipids represent the lipid group that contributed with most ions to this discrimination (Figure 4). It should also be highlighted the increment of lysolipids (LPC, LPE, DGMG, MGMG, and MGTS) in spring samples.

### 2.2. Seasonal Changes in the Fatty Acid Profile

Total fatty acids displayed the same tendency recorded for total lipid extracts, with a significant reduction in spring samples from all three sampling sites when compared to those collected during autumn (Figure 1b). Polyunsaturated fatty acids (PUFA) was the most abundant fatty acid class in all sites and seasons but displaying a decrease in spring when compared to autumn (Table 1). All samples presented a high proportion of *n*-3 PUFA, ranging from 27% to 33% of total fatty acid amount. In wild samples, the most abundant fatty acids showed a decrease in their abundance in spring, although in IMTA samples this tendency was only verified for 16:0, 18:1 *n*-9, 18:2 *n*-6, and 20:4 *n*-6 (Figure 5). IMTA samples showed a seasonal stability in the levels of *n*-3 most abundant fatty acids (Figure 5).

### 2.3. Seasonal Changes in the Pigment Profile

Total lipid extracts of wild *C. tomentosum* samples presented a significantly higher amount of total pigments in autumn when compared to spring. Nevertheless, these differences were not recorded in domesticated samples collected at an IMTA system (Figure 1c). Pigments recorded in *C. tomentosum* from all sampled sites were carotenoids siphonaxanthin (Siph), *trans-* and *cis*-neoxanthin (*t*-Neo and *c*-Neo), violaxanthin (Viola), siphonaxanthin dodecenoate (Siph-do), and β,ε-carotenes (β,ε-Car) and the chlorophylls *a* and *b* (Chl *a* and Chl *b*) (Figure 6). Pigment profiles of wild samples showed a significantly lower amount of all pigments during spring (except for Viola in Aguda samples). In the domesticated samples only three pigments presented significant differences between autumn and spring (e.g., *t*-Neo, *c*-Neo, and Viola) but, unlike wild samples, these pigments were present in higher levels during spring.

### 2.4. Seasonal Changes in the Antioxidant Activity

DPPH assays showed that lipid extracts from samples collected during spring presented a lower inhibition concentration (IC) than that displayed by samples collected during autumn (Table 2). The lower concentration providing 50% of inhibition (IC_50_) was 134.2 ± 46.0 µg mL^−1^ with a Trolox Equivalent (TE) of 115.1 ± 35.2 µmol Trolox g^−1^ of lipid of *C. tomentosum* from Barra–S (Table 2). Results from ABTS assay also revealed a lower IC for spring samples, with the exception of those from Aguda that presented a slightly higher IC in spring than in autumn (Table 2). The lower concentration providing 50% of inhibition (IC_50_) was 22.2 ± 2.2 µg mL^−1^ with a TE of 927.2 ± 92.1 μmol Trolox g^−1^ of lipid from domesticated samples collected in spring (Table 2).

## 3. Discussion

In general, marine organisms exhibit a high plasticity to environmental changes in order to optimize their metabolism and compensate to seasonal variations in their home habitats [9,15,26]. In the present study, spring samples from wild populations presented a lower total amount of lipids than early-autumn samples, but these seasonal differences were not recorded in cultured macroalgae. A similar trend was observed in their fatty acid profiles and total amount of photosynthetic pigments, although the level of fatty acids in spring was significantly lower in all sampled sites. The differences observed between macroalgae grown in their natural habitat and cultured using an IMTA framework are expectably due to fluctuations in their growing environments. While both wild populations were exposed to environmental fluctuations associated with seasonal related changes (e.g., temperature, light, and air exposure), IMTA samples were farmed under relatively controlled conditions throughout the year. Early stages are cultivated from a domesticated strain and in fully controlled laboratory conditions before transfer to outdoor tanks. Outside, some of the IMTA conditions included constant immersion, stable inorganic nutrient supply, controlled water temperature by changing water flow, and management of irradiance levels by the use of shading nets.

Wild macroalgal populations exposed to environmental fluctuations showed a dominant seasonal pattern on lipid content characterized by an increase during winter, likely as a response to shorter days, lower irradiance levels, and temperature [14,15,27,28]. These changes promote a number of adaptations in the algae, such as the enhancement of the surface of thylakoid membranes through the accumulation of glycolipids, located predominantly in photosynthetic membranes [29]. Our dataset on wild *C. tomentosum* populations identified the highest lipid content in samples collected in September (early-autumn), while lower lipid amounts were recorded in May (spring). Sea water temperature in the study areas is usually higher in the early autumn [30], thus revealing that other factor than increased water temperature can influence the content in lipids. The north-western coast of the Iberian Peninsula is influenced by strong upwelling events that transport colder and nutrient-rich water from bottom to surface layers during spring-summer months and raise the net ecosystem production [30,31] (see Appendix A). This increment in nutrient concentrations coincides with high irradiance levels, boosting the growth of wild populations of *C. tomentosum* [22,32]. In this sense, the fertilizing effect of upwelling events experienced by wild populations during summer months may have contributed to the higher total lipid and fatty acid contents displayed by *C. tomentosum* in early autumn. An annual sampling will allow to survey lipid dynamics along seasons and estimate upwelling influence in wild population lipidomes.

The PCA analysis performed with lipid profiles showed a very close grouping in wild samples collected during spring, indicating the importance of environmental conditions in the biochemical composition of *C. tomentosum* from different locations (Aguda and Barra). IMTA samples showed two different groups, associated with wild samples from the opposite season. While wild samples were expose to tidal and more severe seasonal fluctuations, IMTA samples were cultivated in tanks under more stable conditions as referred above. This may also explain the low seasonal variability in the total lipid amount of IMTA samples.

At a molecular level, galactolipids exhibited a major variation. These glycolipids are the main lipid constituent of thylakoid membranes and it has already been described that under low irradiance conditions, such as those occurring during days with a short photoperiod, these lipids increase their relative abundance as an adaptation of the photosynthetic machinery [29]. Considering the contribution of the amount of each lipid class, wild samples did not display any significant differences in their main glycolipid classes in autumn versus spring, with the exception of SQDG and DGDG in Barra samples that showed an increase from autumn to spring. Similar results were obtained by Schmid et al. [28] when analyzing the lipidome of the red macroalgae *Palmaria palmata*. Although, a lower amount of glycolipids in autumn could be related with a degradation of chloroplast membranes [28], this was not verified in the present study due to the significantly higher concentrations of photosynthetic pigments recorded in *C. tometosum* samples from the wild collected in September. Additionally, sample discrimination by season was supported by an increase in normalized peak areas of several lyso forms of galactolipids (MGMG and DGMG) and phospholipids (LPE, LPC, and LPG) in samples collected from the wild during spring. Curiously, wild samples presented an increase on the relative abundance of all lysolipids in spring. However, it is worth considering that in the IMTA samples the trend was the opposite, with a decrease of lysolipids in spring samples. The increase of lysolipids in spring recorded in samples from the wild may be related with a protective mechanism against photodamage caused by an increased in irradiance during spring months. As already confirmed in other photosynthetic organisms, exposure to high irradiance impairs photosynthesis due to an excessive production of reactive oxygen species (ROS) [33]. The accumulation of lysolipids during spring may be related with a mechanism involved in heat or light stress tolerance, as suggested by Rosset et al [34] when studying coral photosynthetic endosymbionts exposed to thermal stress.

It should be highlighted the significant increase in almost all lipid classes recorded in Barra samples from autumn to spring. The population of *C. tomentosum* in Barra beach is small and vulnerable, thus could be more vulnerable under more severe environmental changes.

The main lipids in membranes that enclose photosynthetic machinery (MGDG, DGDG, and SQDG) are characterized by being rich in PUFA [35]. These lipids play a relevant role in photosynthesis by supporting organization of the light harvesting complex, chloroplast photosynthetic performance, increasing membrane fluidity and the electron flow between electron acceptors of Photosystem II [29,36]. Irradiance is an essential physiological mediator in fatty acid regulation in these membranes. Under low irradiance there is an increase of PUFA in thylakoid membranes, whereas under high light conditions PUFA occur in lower proportions [37,38]. The lower levels of PUFA are associated with a decrease of proton leakage, which reduces metabolic costs [39]. In this sense, according to the literature, under low light conditions it is expected that an increase of PUFA, such as 16:3 *n*-3 and 18:3 *n*-3, is associated with galactolipids to increase fluidity and activity of thylakoid membranes [15,28,40]. Furthermore, several authors have suggested a connection between certain fatty acids, such as 18:4 *n*-3 (SDA) and 20:5 *n*-3 (EPA), and Chl *a*, suggesting that under low light conditions the increase of this photosynthetic pigment is accompanied by accumulation of these fatty acids [28,41,42]. The results of the present study denote a positive relation between PUFA and photosynthetic pigments. Furthermore, higher levels of PUFA, mainly *n*-3 PUFA, in autumn versus spring samples, mostly significant in algal samples from the wild, can be associated with physiological adaptations triggered by natural shifts in irradiance [38].

Water temperature is another abiotic parameter that plays a major role on fatty acid regulation of marine organisms. Several studies have reported that low temperatures rise levels of PUFA in cells membranes in order to maintain membrane fluidity [35,43]. Nevertheless, in the present study, a higher proportions of PUFA was observed in September, when sea water temperature is higher than spring [30]. Regarding IMTA samples, PUFA were also higher in autumn, with a significant decrease of 18:2 *n*-6 and 20:4 *n*-6 in spring.

Irradiance is a significant factor influencing the profile of photosynthetic pigments. The significantly lower content in photosynthetic pigments displayed during spring versus autumn for samples of wild algae can be related with the increase of the light period and irradiance levels. Spring samples were collected close to the summer solstice (May), when day length is nearly maximal and irradiance levels are higher than in autumn. Photosynthetic organisms adapt their pigment content to prevailing light conditions, with their concentrations increasing under lower irradiance levels as a compensatory response to lower photon availability [14,38,44]. On the contrary, under higher irradiance, less pigments are required to achieve the same light energy absorption, and photosynthetic organisms reduce pigment synthesis to control metabolic costs [39]. Nevertheless, results from IMTA samples contrast with those from wild algae. While in samples from wild algae all surveyed photosynthetic pigments displayed a reduced abundance in these molecules during spring, samples from IMTA exhibited three pigments with a significant increase in May (*t*-Neo, *c*-Neo and Viola). Furthermore, the high content in Chl *a* and *b* recorded for IMTA samples in both seasons may be linked with the increased availability of dissolved nutrients that favor a higher algal Chl content [14]. The presence of the keto-carotenoid siphonaxanthin and its ester siphonaxanthin dodecenoate, that takes the place of lutein in light harvesting of photosynthetic antenna complexes, is characteristic of siphonous green algae [45]. Ganesan et al. have shown relevant anti-angiogenic and anticancer activities of siphonaxanthin derived from the green macroalgae *C. fragile* [10,46].

Antioxidant scavenging activity of *C. tomentosum* lipid extracts was more efficient against ABTS^●+^ than DPPH^●^ radicals. Although antioxidant activity of this macroalga has been previously tested, these tests were performed with aqueous [13] and ethanol extracts [47], but not in crude organic extracts. Lipid extracts with higher antioxidant activities originated from samples collected in May, suggesting that bioactive components with potential antioxidant properties as radical scavenging agents are present in higher levels in this period. These results must be related with the environmental conditions prevailing before sampling. Spring is a season with higher contrasting conditions than early autumn. Then as a protective mechanism, *C. tomentosum* enriched in antioxidant components, such as glycolipids, during this period. In this sense, studies addressing seasonal shifts in bioactivity of natural products detected maximal bioactivity and antimicrobial potential in the late spring and summer months [14,48,49]. This increment in active compounds during these seasons may constitute a protective mechanism in response to environmental change conditions, such as maximum values of contrasting temperature (day versus night), higher irradiance, and fouling pressure. Nevertheless, a recent lipidomic study addressing *Fucus vesiculosus* harvested in a IMTA system revealed that a greater amount of bioactive components were present in this alga during winter than during spring [15].

Algae present multiple mechanisms of chemical defense against biotic (e.g., pathogen infections, grazers, and epiphytes) and abiotic (e.g., UV radiation, osmotic stress, and desiccation) stressors [50,51]. Intertidal marine algae experience extreme conditions during the tidal cycle, being exposed to intense light, temperature fluctuations, desiccation, and osmotic stress, which ultimately lead to the formation of oxidizing agents, such as free radicals, that could damage their structural components [52]. Over the phylogenetic pathway, marine organisms were able to modify their metabolism and produce an array of compounds to avoid this type of oxidizing damage. Additionally, algae have evolved protective mechanisms against oxidation that promote the production of metabolites to re-establish cell metabolism, which can modify the antioxidant system to correct cellular redox balance [18,52]. Products of algal metabolism have been the target of multiple studies due to their bioactivity [2,4,11,12,46]. Several authors highlight the role of secondary metabolites in the presence of bioactive compounds, such as phenolics [4,13]. Nevertheless, some compounds flagged by researchers due to their bioactivity are paramount in algal metabolism, such as glycolipids, pigments, or PUFAS [1,2,10,11,12,46]. Total lipid extracts contain several compounds with putative bioactivity, such as glycolipids and fatty acids (e.g., linoleic acid, C18:2 *n*-6), oxylipins, or pigments [10,46,53]. Pigments are recognized by their bioactivity as well, namely carotenoids which are well-known by their antioxidative capacity [1,54]. Carotenoids serve as accessory pigments to harvest light for photosynthesis and act as photoprotectors under high light stress, playing a protective role against photooxidative damage [55]. Both fatty acid and pigment profiles have been identified by their antioxidant capacity, such as carotenoids esterified with oleic acid [56], chlorophylls, and their derivatives. Several compounds can contribute individually or in synergy to the antioxidant activity of lipid extracts from *C. tomentosum*. Additionally, it must be expressed that *C. tomentosum* belongs to the Bryopsidales order, where photoprotective mechanisms are unclear and require further investigations [57,58].

## 4. Materials and Methods

### 4.1. Reagents

Phospholipid standards 1,2-dimyristoyl-sn-glycero-3-phosphocholine (dMPC), 1,2-dimyristoyl-snglycero-3-phosphoethanolamine (dMPE), 1,2-dipalmitoyl-sn-glycero-3-phosphatidylinositol (dPPI), 1,2-dimyristoyl-sn-glycero-3-phospho-(10-rac-)glycerol (dMPG), 1,2-dimyristoyl-sn-glycero-3-phospho-l-serine (dMPS), 10,30-bis(1–dimyristoyl-sn-glycero-3-phospho)-glycerol (tMCL), 1,2-dimyristoyl-sn-glycero-3-phosphate (dMPA), *N*-heptadecanoyl-d-erythro-sphingosylphosphorylcholine (SM d18:1/17:0) , and 1-nonadecanoyl-2-hydroxy-sn-glycero-3-phosphocholine (LPC) were purchased from Avanti Polar Lipids, Inc. (Alabaster, AL, USA).

Chloroform (CHCl3), methanol (MeOH), ethanol, and acetonitrile were purchased from Fisher scientific (Leicestershire, UK); all the solvents were of high-performance liquid chromatography (HPLC) grade and were used without further purification. DPPH^●^ was purchased from Aldrich (Milwaukee, WI). 2,20-Azino-bis(3-ethylbenzothiazoline-6-sulfonic acid) diammonium salt (ABTS^●+^) was obtained from Fluka (Buchs, Switzerland). Ammonium acetate and 6-hydroxy-2,5,7,8-tetramethylchromane-2-carboxylic acid (Trolox) were purchased from Sigma-Aldrich (St. Louis, MO, USA). All the other reagents and chemicals used were of the highest grade of purity commercially available. Milli-Q water was used as ultrapure water (Synergysup®, Millipore Corporation, Billerica, MA, USA).

### 4.2. Sampling

*Codium tomentosum* samples were collected in three different locations of the Portuguese coast, Aguda beach (Aguda samples) (Vila Nova de Gaia, 41°02′52.8″ N, 8°39′14.0″ W), Paredão da Barra beach (Barra samples) (Ilhavo, 40°38′24.1″ N 8°44′57.0″ W), and on a land-based integrated multitrophic aquaculture (IMTA) in Aveiro (ALGAplus, Ilhavo, 40°36′43.9″ N 8°40′26.2″ W) (IMTA samples). Samples were collected along two different periods Autumn 2017 (September 2017 for Aguda and Barra samples/October 2017 for IMTA samples) and Spring 2018 (all samples collected during May 2018). ALGAplus farms seaweeds integrated with fin fish production (seabream and seabass) using an IMTA system. Five samples (*n* = 5) were used for each local and season (Total samples: 5 replicates × 3 locals × 2 seasons = 30 samples). After collection, *C. tomentosum* samples were rinsed with distilled freshwater, cleaned to remove epiphytic organisms, frozen, freeze-dried, and stored individually at −80 °C for biochemical analysis.

### 4.3. Lipid Extraction

The Bligh and Dyer method [59] was modified to extract total lipids from *C. tomentosum* samples. A total biomass of 250 mg of macroalga was mixed with 2.5 mL of methanol and 1.25 mL of chloroform in glass centrifuge tubes. Following vigorous homogenization and a small step of sonication (1 min), samples were incubated on ice for 2 h and 30 min on an orbital shaker (Stuart equipment, Bibby Scientific, Stone, UK). Samples were then centrifuged at 626 *g* for 10 min at room temperature and the organic phase, containing the lipids, was collected. A re-extraction step was performed in the original biomass, and the resulting organic phase was combined with the first. Ultrapure water was added to the organic phase in order to resolve a two-phase system. Samples were centrifuged at 626 *g* for 10 min at room temperature, and the lipid-containing organic lower phase was collected. The upper aqueous phase containing no lipid compounds was discharged. Lipid extracts, recovered from the organic phases, were dried under a nitrogen stream and preserved at −20 °C for further analysis [20]. The final weight of lipid extracts was determined by gravimetry.

### 4.4. Hydrophilic Interaction Liquid Chromatography–Mass Spectrometry (HILIC–LC–MS)

Total lipid extracts were analyzed by hydrophilic interaction liquid chromatography on a HPLC Ultimate 3000 Dionex (Thermo Fisher Scientific, Bremen, Germany) with an autosampler coupled online to a Q-Exactive hybrid quadrupole mass spectrometer (Thermo Fisher, Scientific, Bremen, Germany). The analyses performed involved the use of a solvent system consisting of two mobile phases. Mobile phase A consisted of water, acetonitrile and methanol (25%, 50%, and 25%), with 2.5 mM ammonium acetate, and mobile phase B was a mixture of acetonitrile and methanol (60% and 40%), with 2.5 mM ammonium acetate. Elution started with 10% of mobile phase A, which was held isocratically for 2 min, followed by a linear increase to 90% of mobile phase A within 13 min and maintained for 2 min. After this procedure, conditions were returned to the initial settings in 13 min (3 min to decrease to 10% of phase A and a re-equilibration period of 10 min prior next injection). To perform HILIC–LC–MS analyses, 10 μg of total lipid extract, 2 µL of phospholipid standards mix (dMPC—0.01 µg, dMPE—0.01 µg, LPC—0.01 µg, dPPI—0.04 µg, dMPG—0.006 µg, dMPS—0.02 µg, tMCL—0.04 µg, SM(17:0/d18:1)—0.01 µg, and dMPA—0.04 µg) and 88 µL of eluent (10% of mobile phase A and 90% of mobile phase B) were mixed and a 5 µL of this mixture was injected into the Ascentis Si column HPLC Pore column (10 cm × 1 mm, 3 µm, Sigma-Aldrich), with a flow rate of 50 µL min^−1^ at 35 °C. Acquisition in the Orbitrap^®^ mass spectrometer was performed in both positive (electrospray voltage 3.0 kV) and negative (electrospray voltage −2.7 kV) modes, with high resolution with 70,000 and automatic gain control (AGC) target of 1 × 10^6^. Capillary temperature was 250 °C, and the sheath gas flow was 15 U. For MS/MS determinations, a resolution of 17,500 and AGC target of 1 × 10^5^ was used, and the cycles consisted in one full scan mass spectrum, and ten data-dependent MS/MS scans were repeated continuously throughout the experiments, with the dynamic exclusion of 60 s and an intensity threshold of 2 × 10^4^. Normalized collision energy™ (CE) ranged between 20, 25, and 30 eV. Data acquisition was performed using the Xcalibur data system (V3.3, Thermo Fisher Scientific, USA). Two analytical replicas were acquired by sample.

### 4.5. Fatty Acid Analysis Using Gas Chromatography-Mass Spectrometry (GC-MS)

Fatty acids of total lipid extract were transmethylated to be analyzed by GC-MS. In order to obtain fatty acid methyl esters (FAMEs). A 30 μg of lipid was transferred to a Pyrex glass tube and dried under nitrogen. Dry lipids were mixed with 1 mL of *n*-hexane containing a C19:0 internal standard (1 μg mL^−1^, CAS number 1731-94-8, Merck, Darmstadt, Germany) and 200 μL of a methanolic solution of potassium hydroxide (2 mol L^−1^) and vortexed for 2 min. A 2 mL of a saturated solution of sodium chloride was added and the sample was centrifuged at 626× *g* for 10 min. The organic phase containing the FAMEs was transferred to a microtube and dried under nitrogen [60]. FAMEs were then dissolved in 60 µL *n*-hexane and 2 μL of this solution was used for GC-MS analysis on an Agilent Technologies 6890 N Network chromatograph (Santa Clara, CA, USA) equipped with a DB-FFAP column with 30 m length, an internal diameter of 0.32 mm and a film thickness of 0.25 μm (J&W Scientific, Folsom, CA, USA). The GC was connected to an Agilent 5973 Network Mass Selective Detector operating with an electron impact mode at 70 eV and scanning the mass range *m/z* 50–550 in a 1 s cycle in a full scan mode acquisition. The oven temperature was programmed from an initial temperature of 80 °C, standing at this temperature for 3 min, followed by three consecutive linear increments to 160 °C at 25 °C min^−1^, to 210 °C at 2 °C min^−1^, and to 250 °C at 30 °C min^−1^. Temperature was maintained at 250 °C for 10 min. The injector and detector temperatures were 220 and 250 °C, respectively. Helium was used as carrier gas at a flow rate of 1.4 mL min^−1^ [61]. Fatty acid amounts were calculated using calibration curves of FAME standards Supelco 37 Component FAME Mix (ref. 47885-U, Sigma-Aldrich, Darmstadt, Germany) analyzed by GC–MS under the same conditions of FAMEs samples. Fatty acid identification was performed by comparing their retention time and mass spectrum with MS spectra of the commercial FAME standards Supelco 37 and confirmed by interpretation of MS spectra [62].

### 4.6. Pigments Analysis Using HPLC

Lipid extracts (500 µg) were dissolved in 1 mL of 95% cold buffered methanol (2% ammonium acetate). Samples were sonicated for 30 s and briefly vortexed to ensure complete resuspension. Extracts were filtered through 0.2-µm PTFE membrane filters and immediately injected into a HPLC system (Shimadzu, Kyoto, Japan) with a photodiode array detector. Chromatographic separation was carried out using a Supelcosil C18 column (25 cm length; 4.6 mm diameter; 5 µm particles; Sigma-Aldrich, St. Louis, MO, USA) for reverse phase chromatography and a 35 min elution program. The solvent gradient followed Kraay et al. [63] with an injection volume of 50 µL and a flow rate of 0.6 mL min^−1^. Pigments were identified from absorbance spectra and retention times and concentrations were calculated in comparison with pure crystalline standards (DHI, Hørsolm, Denmark).

### 4.7. Antioxidant Assays

#### 4.7.1. 2-Diphenyl-1-Picrylhydrazyl Radical Assay—DPPH Radical Scavenging Activity

The antioxidant scavenging activity against the α,α-diphenyl-β-picrylhydrazyl radical (DPPH^●^) was evaluated using a previously described method [64] applied with some modifications. A stock solution of DPPH^●^ in ethanol (250 µmol L^−1^) was prepared and diluted to provide a working solution with an absorbance value of ~0.9 measured at 517 nm using a UV-vis spectrophotometer (Multiskan GO 1.00.38, Thermo Scientific, Hudson, NH, USA) controlled by the SkanIT software version 3.2 (Thermo Scientific). To evaluate the radical stability, a volume of 150 µL of ethanol was added to 15 microplate wells followed by addition of 150 µL of DPPH^●^ diluted solution during an incubation period of 120 min at room temperature, with absorbance measured at 517 nm every 5 min. For evaluation of the radical scavenging potential, a volume of 150 µL of *C. tomentosum* lipid extract (50, 100, 200, 500 µg mL^−1^ in ethanol) and 150 µL of Trolox standard solution (10, 25, 50, and 75 µg mL^−1^ in ethanol) were placed in each well followed by addition of 150 µL of DPPH^●^ diluted solution, and again an incubation period of 120 min, with absorbance measured at 517 nm every 5 min. Control lipid extracts were tested by replacing 150 µL of DPPH^●^ diluted solution by 150 µL of ethanol. Radical reduction was monitored by measuring the decrease in absorbance during the reaction, thereby quantifying radical scavenging, which is accompanied by a radical color change. All measurements were performed in triplicate on two different days. The % of DPPH radical remaining was determined according to: % DPPH remaining = ((Abs samples − Abs control) after 120 min/(Abs samples − Abs control) at the beginning of reaction) × 100.

The free radical-scavenging activity of samples was determined as the percentage of inhibition of DPPH radical according to: Inhibition% = ((Abs DPPH − (Abs samples − Abs control))/Abs DPPH) × 100.

The concentration of samples capable of reducing 50% of DPPH radical after 120 min (IC_50_) were calculated by linear regression using the concentration of samples and the percentage of the inhibition curve. The activity is expressed as Trolox Equivalents (TE, µmol Trolox g^-1^ of lipid extract), according to:

TE = IC_50_ Trolox (µmol L^−1^) × 1000 /IC_50_ of samples (µg mL^−1^).

#### 4.7.2. 2,20-Azino-bis-3-Ethylbenzothiazoline-6-Sulfonic Acid Radical Cation Assay—ABTS Radical Scavenging Activity

The antioxidant scavenging activity against the 2,20-azino-bis-3-ethylbenzothiazoline-6-sulfonic acid radical cation (ABTS^●+^) was evaluated using a previously described method [64,65] applied with some modifications. The ABTS radical solution (3.5 mmol L^−1^) was prepared by mixing 10 mL of ABTS stock solution (7 mmol L^−1^ in water) with 10 mL of potassium persulfate K2S2O8 (2.45 mmol L^−1^ in water) [65,66]. This mixture was kept for 12 h at room temperature and was diluted in ethanol to obtain an absorbance value of ~0.9 measured at 734 nm using a UV–-vis spectrophotometer (Multiskan GO 1.00.38, Thermo Scientific, Hudson, NH, USA) controlled by the SkanIT software version 3.2 (Thermo Scientific). For an evaluation of the radical stability, a volume of 150 µL of ethanol was added to 15 microplate wells followed by addition of 150 µL of ABTS^●+^ diluted solution and an incubation period of 120 min, with absorbance measured at 734 nm every 5 min. For an evaluation of the radical scavenging potential, a volume of 150 µL of lipid extract (50, 100, 200, and 500 µg mL^−1^ in ethanol) and 150 µL of Trolox standard solution (4, 8, 16, 28, 40, and 56 µmol L^−1^ in ethanol) were placed in each well followed by addition of 150 µL of ABTS^●+^ diluted solution, and a new incubation period of 120 min, with absorbance measurements at 734 nm every 5 min. The control lipid extracts were also assayed by replacing 150 µL of ABTS^●+^ diluted solution by 150 µL of ethanol. Radical reduction was monitored by measuring the decrease in absorbance during the reaction, thereby quantifying radical scavenging, which is accompanied by a radical color change. The percentage of ABTS radical remaining was determined according to: % ABTS remaining = ((Abs samples − Abs control) after 120 min/(Abs samples − Abs control) at the beginning of reaction) × 100.

The free radical-scavenging activity of samples was determined as the percentage of inhibition of ABTS radical according to: Inhibition% = ((Abs ABTS − (Abs samples − Abs control))/Abs ABTS) × 100.

The concentration of samples capable of reducing 50% of ABTS radical after 120 min (IC_50_) were calculated by linear regression using the concentration of samples and the percentage of the inhibition curve. The activity is expressed as Trolox Equivalents (TE, µmol Trolox g^−1^ of lipid extract), according to: TE = IC_50_ Trolox (µmol L^−1^) × 1000/IC_50_ of samples (µg mL^−1^)

### 4.8. Data Analysis

For lipidomic analysis, polar lipid molecular species were identified by the *m/z* ratio of the ions observed in LC-MS spectra. Ion peak integration and assignments were performed using MZmine version 2.42 [67], based on an in-house lipid database. This software allows for filtering and smoothing, peak detection, peak processing, and assignment. During the processing of the raw data acquired in full MS mode, all the peaks with raw intensity lower than 1 × 10^4^ were excluded. Identification of polar lipid molecular species was based on the molecular ions observed in the LC-MS spectra, typical retention time and mass accuracy (Qual Browser) of ≤ 5 ppm (Appendix A). Quantitation of lipid species was performed by exporting integrated peak areas in a data matrix using the comma separated values (.csv) format. Data normalization was performed by dividing the peak areas of the extracted ion chromatograms (XIC) of each lipid molecular species of each class by the peak area of the internal standard selected for the class.

### 4.9. Statistical Analysis

One-way ANOVAs were performed to compare total lipid extract, fatty acid and pigment contents in autumn versus spring for each sampling site. The same test was used to compare normalized peak areas of lipid classes, fatty acid and pigment profiles. Assumptions of normality and homogeneity of variance were verified prior to analysis through Shapiro-Wilks and Levene’ s tests, respectively. Whenever these assumptions were not verified, Kruskal-Wallis test was employed. These statistical analyses were performed using R 3.6.0 [68] with a level of statistical significance of *p* < 0.05.

Lipidomic data assessment was performed using Metaboanalyst [69]. Variables equal to 0 were replaced by a small value (half of the minimum positive value in the original data) assuming to be the detection limit. A data filtering process was performed in order to remove variables that show low repeatability, using the relative standard deviation (RSD = SD/mean). Normalized data were glog transformed and auto-scaled. Principal component analysis (PCA) was performed to visualize the general 2D clustering of *C. tomentosum* samples from three different locations and collected at two different seasons. In order to identify the molecular species that more contribute to the differences between groups, heat maps were performed using Euclidean distance measure and Ward clustering algorithm and the top 25 variables were ranking using ANOVA test.

All experimental data are shown as mean ± standard deviation (*n* = 5).

## 5. Conclusions

The present study demonstrated the existence of spatial and seasonal adaptations in lipid and photosynthetic pigment profiles, as well as in the antioxidant activity of *C. tomentosum*, highlighting the relevant role of ecological and environmental conditions in the metabolism of biological samples. These results emphasize the need for a comprehensive understanding of natural plasticity in the metabolism of marine organisms, which must be taken into account when conducting analytical and prospective studies. Understanding the seasonal influence in biochemical composition of organisms with potential in biotechnological applications is crucial to maximize the production of high-value compounds under controlled conditions, mimicking nature.

## Figures and Tables

**Figure 1 marinedrugs-18-00188-f001:**
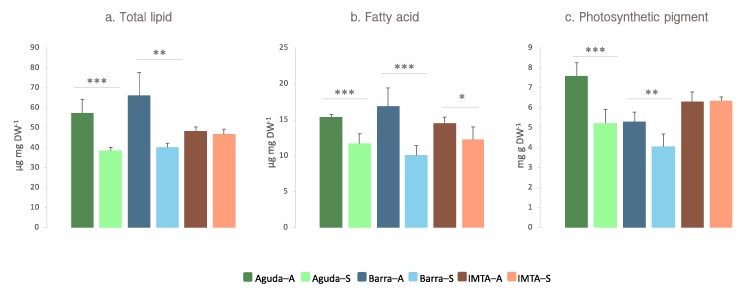
(**a**) Total lipid, (**b**) fatty acid, and (**c**) photosynthetic pigment content in lipid extracts of *Codium tomentosum* samples from three different sites (Aguda, Barra, and IMTA) and two seasons (autumn and spring). Differences between sampling seasons in the same location were determined by one-way ANOVA, * *p* < 0.05, ** *p* < 0.01, *** *p* < 0.001. Abbreviations: Aguda–A: Aguda Autumn; Aguda–S: Aguda Spring; Barra–A: Barra Autumn; Barra–S: Barra Spring; IMTA–A: IMTA Autumn; IMTA–S: IMTA Spring.

**Figure 2 marinedrugs-18-00188-f002:**
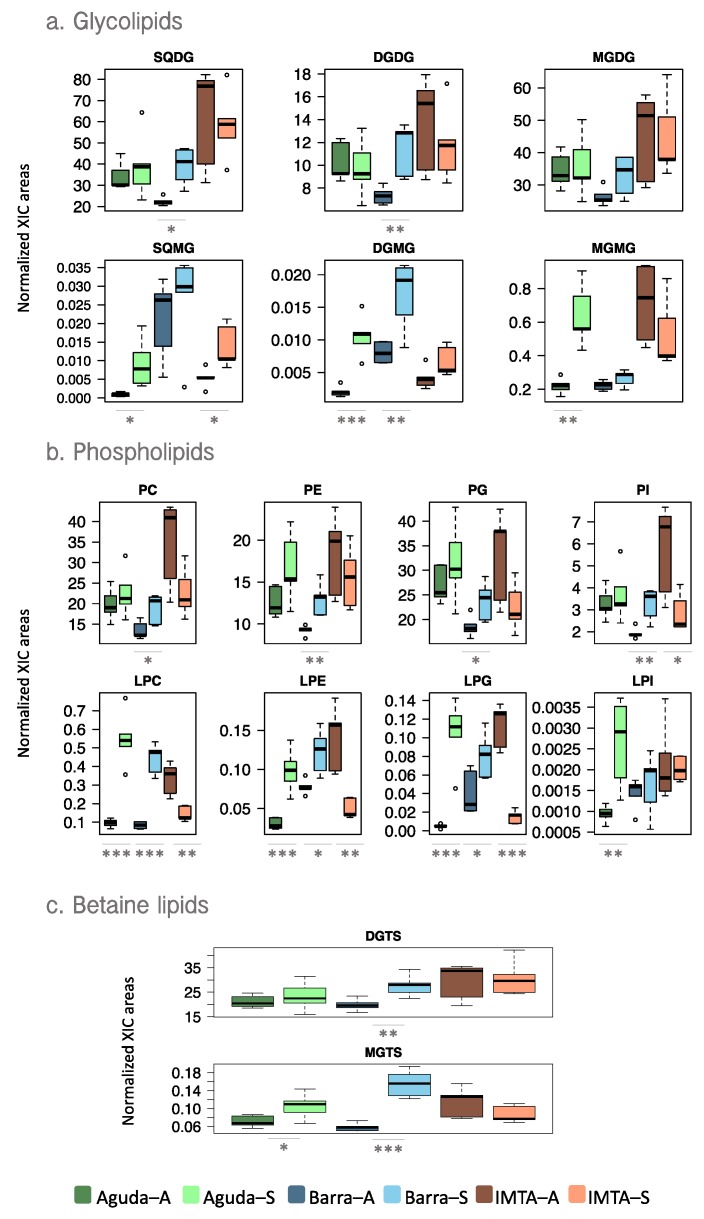
Box plots of normalized peak areas of (**a**) glycolipid, (**b**) phospholipid, and (**c**) betaine lipid classes in total lipid extracts of *Codium tomentosum* samples from three different sites (Aguda, Barra, and IMTA) and two seasons (autumn and spring). Differences between sampling seasons in the same site were determined by one-way ANOVA, * *p* < 0.05, ** *p* < 0.01, *** *p* < 0.001. Abbreviations: Aguda–A: Aguda Autumn; Aguda–S: Aguda Spring; Barra–A: Barra Autumn; Barra–S: Barra Spring; IMTA–A: IMTA Autumn; IMTA– S: IMTA Spring.

**Figure 3 marinedrugs-18-00188-f003:**
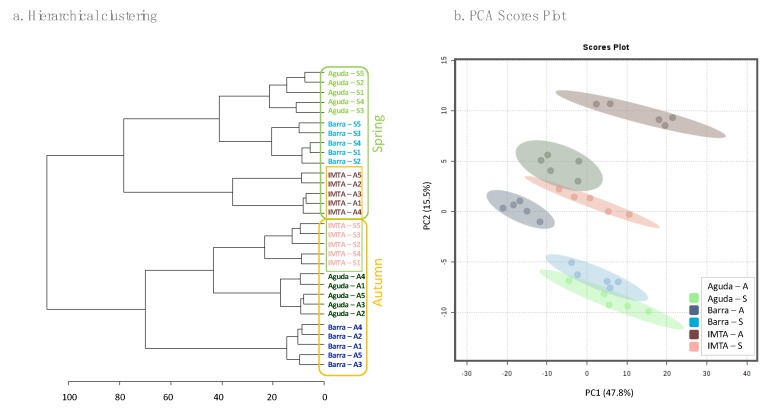
(**a**) Hierarchical clustering analysis and (**b**) principal component analyses (PCA) scores plot using normalized extracted ion chromatograms (XIC) areas of all molecular ions identified in *Codium tomentosum* samples from three different sites (Aguda, Barra and IMTA) and two seasons (autumn and spring). In (**a**) yellow boxes present samples collected in autumn and green boxes samples collected in spring, shaded boxes correspond with IMTA samples. Abbreviations: Aguda–A: Aguda Autumn; Aguda–S: Aguda Spring; Barra–A: Barra Autumn; Barra–S: Barra Spring; IMTA–A: IMTA Autumn; IMTA–S: IMTA Spring.

**Figure 4 marinedrugs-18-00188-f004:**
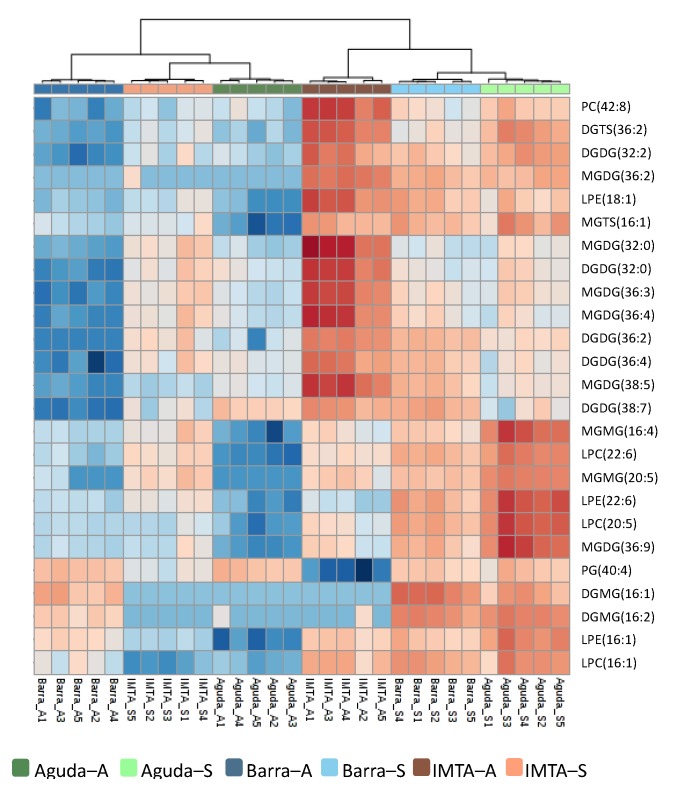
Two-dimensional hierarchical clustering heat map of the top 25 most significant molecular ions discriminating sampling seasons. Levels of relative abundance are shown on the color scale and numbers indicate the fold difference from the mean. Dendrogram in the top represents the clustering of the sample groups. Lipid species are labelled as follows: AAAA(C:N) (AAAA = lipid class; *C* = total of carbon atoms in fatty acid(s); *N* = number of double bonds). Abbreviations: Aguda–A: Aguda Autumn; Aguda–S: Aguda Spring; Barra–A: Barra Autumn; Barra–S: Barra Spring; IMTA–A: IMTA Autumn; IMTA–S: IMTA Spring.

**Figure 5 marinedrugs-18-00188-f005:**
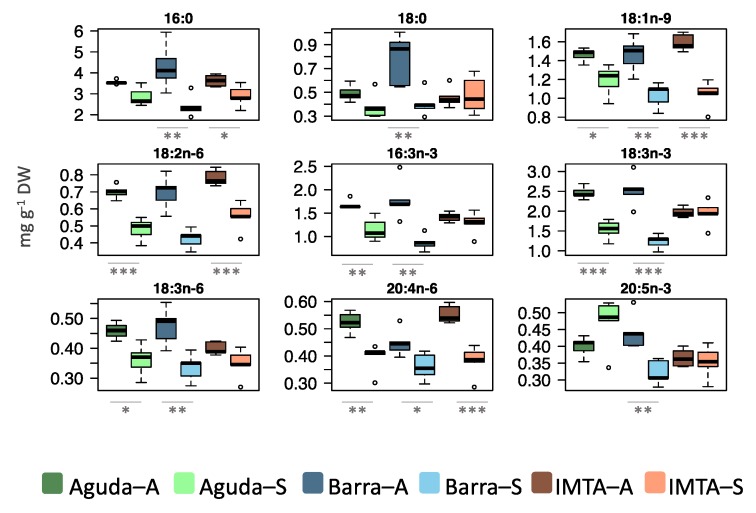
Box plots of the most abundant fatty acids in lipid extracts of *Codium tomentosum* samples. Differences between sampling seasons in the same site were determined by one-way ANOVA, * *p* < 0.05, ** *p* < 0.01, *** *p* < 0.001. Abbreviations: Aguda–A: Aguda Autumn; Aguda–S: Aguda Spring; Barra–A: Barra Autumn; Barra–S: Barra Spring; IMTA–A: IMTA Autumn; IMTA–S: IMTA Spring.

**Figure 6 marinedrugs-18-00188-f006:**
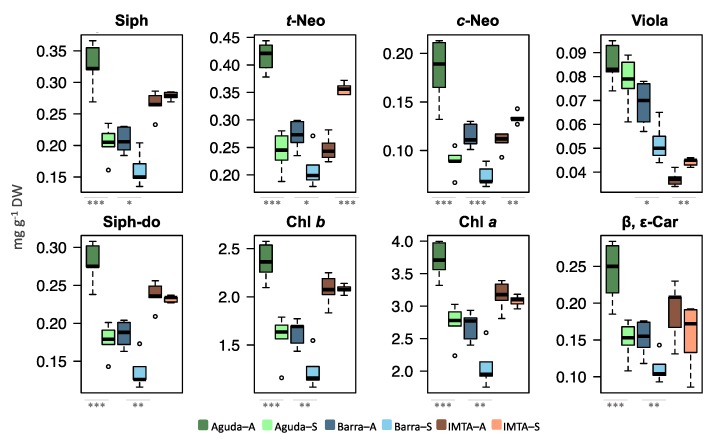
Box plots of photosynthetic pigments identified in lipid extracts of *Codium tomentosum* samples. Differences between sampling seasons in the same site were determined by one-way ANOVA, * *p* < 0.05, ** *p* < 0.01, *** *p* < 0.001. Abbreviations: Aguda–A: Aguda Autumn; Aguda–S: Aguda Spring; Barra–A: Barra Autumn; Barra–S: Barra Spring; IMTA–A: IMTA Autumn; IMTA–S: IMTA Spring.

**Table 1 marinedrugs-18-00188-t001:** Fatty acid profiles of total lipid extracts of *Codium tomentosum* samples from three different sites (Aguda, Barra, and IMTA) and two seasons (autumn and spring). Data of individual fatty acids are given in mg g^−1^ DW. Data are presented as mean ± SD (*n* = 5).

	Aguda–A	Aguda–S	Barra–A	Barra–S	IMTA–A	IMTA–S
12:0	0.31 ± 0.04	0.22 ± 0.02	0.33 ± 0.05	0.22 ± 0.02	0.26 ± 0.01	0.24 ± 0.02
14:0	0.35 ± 0.03	0.28 ± 0.03	0.42 ± 0.05	0.26 ± 0.03	0.33 ± 0.01	0.29 ± 0.03
15:0	0.15 ± 0.02	0.11 ± 0.01	0.17 ± 0.03	0.11 ± 0.01	0.13 ± 0.01	0.12 ± 0.01
16:0	3.54 ± 0.10	2.86 ± 0.45	4.30 ± 1.09	2.41 ± 0.52	3.63 ± 0.28	2.90 ± 0.50
18:0	0.49 ± 0.07	0.38 ± 0.11	0.78 ± 0.21	0.41 ± 0.11	0.46 ± 0.09	0.48 ± 0.16
20:0	0.20 ± 0.02	0.14 ± 0.01	0.24 ± 0.04	0.14 ± 0.01	0.17 ± 0.01	0.17 ± 0.01
22:0	0.36 ± 0.02	0.23 ± 0.04	0.33 ± 0.07	0.24 ± 0.02	0.25 ± 0.02	0.27 ± 0.02
24:0	0.25 ± 0.02	0.17 ± 0.02	0.25 ± 0.05	0.18 ± 0.01	0.21 ± 0.01	0.22 ± 0.01
∑ SFA	5.66 ± 0.06	4.39 ± 0.42	6.82 ± 1.26	3.97 ± 0.65	5.44 ± 0.39	4.69 ± 0.69
16:1 *n*-9	0.19 ± 0.01	0.14 ± 0.01	0.19 ± 0.02	0.14 ± 0.01	0.18 ± 0.01	0.14 ± 0.01
16:1 *n*-7	0.25 ± 0.01	0.32 ± 0.05	0.27 ± 0.04	0.22 ± 0.03	0.25 ± 0.01	0.20 ± 0.03
16:1	0.28 ± 0.02	0.19 ± 0.02	0.26 ± 0.03	0.15 ± 0.02	0.16 ± 0.01	0.18 ± 0.02
18:1 *n*-9	1.46 ± 0.07	1.19 ± 0.16	1.46 ± 0.18	1.03 ± 0.13	1.59 ± 0.09	1.04 ± 0.15
18:1	0.32 ± 0.03	0.28 ± 0.02	0.33 ± 0.05	0.25 ± 0.01	0.30 ± 0.01	0.24 ± 0.00
∑ MUFA	2.50 ± 0.09	2.13 ± 0.26	2.52 ± 0.30	1.78 ± 0.19	2.48 ± 0.12	1.80 ± 0.20
16:2 *n*-6	0.20 ± 0.01	0.14 ± 0.02	0.18 ± 0.03	0.13 ± 0.02	0.27 ± 0.01	0.14 ± 0.02
16:3 *n*-4	0.00 ± 0.00	0.00 ± 0.00	0.12 ± 0.02	0.08 ± 0.01	0.10 ± 0.00	0.09 ± 0.01
16:3 *n*-3	1.68 ± 0.10	1.15 ± 0.24	1.79 ± 0.42	0.87 ± 0.17	1.42 ± 0.10	1.29 ± 0.25
16:4 *n*-1	0.09 ± 0.01	0.11 ± 0.01	0.12 ± 0.02	0.08 ± 0.01	0.09 ± 0.00	0.08 ± 0.01
18:2 *n*-6	0.70 ± 0.04	0.48 ± 0.07	0.70 ± 0.10	0.42 ± 0.06	0.78 ± 0.05	0.56 ± 0.08
18:3 *n*-6	0.46 ± 0.03	0.36 ± 0.05	0.47 ± 0.06	0.34 ± 0.05	0.40 ± 0.02	0.35 ± 0.05
18:3 *n*-3	2.46 ± 0.16	1.54 ± 0.24	2.53 ± 0.40	1.24 ± 0.18	1.97 ± 0.13	1.95 ± 0.33
18:4 *n*-3	0.34 ± 0.02	0.32 ± 0.05	0.39 ± 0.05	0.25 ± 0.03	0.29 ± 0.01	0.29 ± 0.04
20:3 *n*-6	0.22 ± 0.03	0.12 ± 0.03	0.20 ± 0.03	0.14 ± 0.04	0.22 ± 0.02	0.19 ± 0.05
20:4 *n*-6	0.52 ± 0.04	0.39 ± 0.05	0.45 ± 0.05	0.36 ± 0.05	0.55 ± 0.03	0.38 ± 0.06
20:4 *n*-3	0.12 ± 0.01	0.07 ± 0.00	0.13 ± 0.02	0.07 ± 0.00	0.12 ± 0.00	0.09 ± 0.01
20:5 *n*-3	0.40 ± 0.03	0.47 ± 0.08	0.44 ± 0.05	0.32 ± 0.04	0.37 ± 0.03	0.35 ± 0.05
∑ PUFA	7.20 ± 0.36	5.16 ± 0.76	7.52 ± 1.07	4.31 ± 0.58	6.58 ± 0.38	5.75 ± 0.91
∑ *n*-3	5.00 ± 0.28	3.55 ± 0.57	5.28 ± 0.87	2.75 ± 0.39	4.16 ± 0.26	3.97 ± 0.67
∑ *n*-6	2.11 ± 0.09	1.50 ± 0.18	1.99 ± 0.23	1.40 ± 0.18	2.23 ± 0.12	1.62 ± 0.24
TFA	15.36 ± 0.41	11.69 ± 1.38	16.86 ± 2.58	10.05 ± 1.37	14.51 ± 0.87	12.24 ± 1.77

Abbreviations: Aguda–A: Aguda Autumn; Aguda–S: Aguda Spring; Barra–A: Barra Autumn; Barra–S: Barra Spring; IMTA–A: IMTA Autumn; IMTA–S: IMTA Spring; SFA: Saturated fatty acids; MUFA: Monounsaturated FA; PUFA: Polyunsaturated FA; TFA: Total FA.

**Table 2 marinedrugs-18-00188-t002:** Inhibition concentration (IC) of lipid extracts from *Codium tomentosum* providing 50% of inhibition (IC_50_) after 120 min of DPPH and ABTS radical scavenging activity and the corresponding Trolox Equivalent (TE). Samples were collected in three different locations (Aguda, Barra, and IMTA) and in two different seasons (autumn and spring). Data are presented as mean ± SD (*n* = 5).

		DPPH ASSAY		ABTS ASSAY	
		IC50 (µg mL^−1^)	TE (µmol Trolox g^−1^ Lipid Extract)	IC50 (µg mL^−1^)	TE (µmol Trolox g^−1^ Lipid Extract)
AUTUMN	Aguda	249.90 ± 8.04	60.73 ± 1.99	26.34 ± 1.05	828.26 ± 33.04
	Barra	199.55 ± 59.70	81.40 ± 27.33	41.44 ± 5.39	532.22 ± 72.68
	IMTA	209.71 ± 59.63	77.08 ± 25.26	26.24 ± 4.73	850.67 ± 168.96
SPRING	Aguda	184.16 ± 4.94	78.15 ± 2.07	27.40 ± 3.44	753.93 ± 94.55
	Barra	134.22 ± 46.04	115.14 ± 35.20	31.59 ± 5.47	661.65 ± 111.58
	IMTA	139.65 ± 21.93	104.80 ± 17.11	22.22 ± 2.21	927.15 ± 92.09

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
