# Peer review of "Domesticated Populations of Codium tomentosum Display Lipid Extracts with Lower Seasonal Shifts than Conspecifics from the Wild—Relevance for Biotechnological Applications of this Green Seaweed"

_marinedrugs, 2020, doi:10.3390/md18040188_

Round 1

Reviewer 1 Report

In this work, the authors studied the lipidic variation of a common macroalga Codium tomentosum from the northern Atlantic. They use collections in three sites and two seasons and compared the profiles with inland culture of this species. Even though conclusions on the temporal variation seem difficult to raise with only two points the study is exhaustive in the characterization of the lipids but also pigments. It, therefore, deserves publication after addressing the minor comments.

I would recommend the authors to avoid excess self-citations as I noted 7 cited references from the authors not really related to this work.

L30: some metabolites might be common between algae and plants

L55: some secondary metabolites may always be present not only as a response of a stress

L85 the authors use a lot the field of chemical ecology but usually this field involves chemical cues and this is not the case here. I would just use chemical variability.

L90: I am surprised that polar lipids require HILIC column for their analysis. I would think that a common C18 should suffice.

L103 allowed the identification

L396: authors mention an organic phase but the sample was still wet? Is there any water?

Author Response

Reviewer #1

In this work, the authors studied the lipidic variation of a common macroalga Codium tomentosum from the northern Atlantic. They use collections in three sites and two seasons and compared the profiles with inland culture of this species. Even though conclusions on the temporal variation seem difficult to raise with only two points the study is exhaustive in the characterization of the lipids but also pigments. It, therefore, deserves publication after addressing the minor comments.

R1C1 - I would recommend the authors to avoid excess self-citations as I noted 7 cited references from the authors not really related to this work.

Authors’ reply: We have reduced the self-citations. The self-citations in the revised version are related with lipidomic of macroalgae, which help to discuss the results of the present study.

R1C2 - L30: some metabolites might be common between algae and plants

Authors’ replyWe think that Reviewer # 1 is referring L50 in this comment. We have rephased this sentence. Now, it reads:

“Edible seaweeds are known to be rich in bioactive metabolites, some of them not present in land plants” (Line 49 in tack-changes manuscript)

R1C3 - L55: some secondary metabolites may always be present not only as a response of a stress

Authors’ reply: We have rephased this sentence. Now, it reads:

“Bioactivity has been identified in compounds that are essential for macroalgae metabolism, such as lipids and pigments [10–12], but also in products originating from their secondary metabolism [4,13].” (Line 53 in tack-changes manuscript)

R1C4 - L85 the authors use a lot the field of chemical ecology but usually this field involves chemical cues and this is not the case here. I would just use chemical variability.

Authors’ reply: We have removed the term “chemical ecology” and replaced it by “marine chemistry” (Line 69 in tack-changes manuscript) or “biochemical” (Line 85 in tack-changes manuscript) according with the meaning of the sentence.

R1C5 - L90: I am surprised that polar lipids require HILIC column for their analysis. I would think that a common C18 should suffice.

Authors’ reply: We understand the comment of the Reviewer #1 and appreciate the suggestion. We currently use HILIC for lipidomic studies. In lipidomic studies normal phase (NP) or HILIC columns and reversed-phase C18 columns have been used. The separation of lipids in both cases are based on different properties: NP or HILIC columns allow the separation of lipids based on their hydrophilicity, which is mainly dependent on the polar head properties, allowing the separation of different lipid classes in a single run. In reversed-phase C18 columns lipids elute based on their hydrophobic properties, and thus separate the lipid species based on number of carbons and degree of unsaturation of fatty acyl substituents, and thus there are co-elution of lipid species of different classes. This is not very good since, for example, in the case of PC and PE classes, there are some molecular species that can have the same molecular weight, then they co-elute in the reverse phase columns, and their peaks can be overlapped and impairing their proper identification and quantification.

R1C6 - L103 allowed the identification

Authors’ reply: Corrected as suggested. Now, it reads:

“allowed the identification of 296 molecular ions” (Line 107 in tack-changes manuscript)

R1C7 - L396: authors mention an organic phase but the sample was still wet? Is there any water?

Authors’ reply: As it was referred in the section 4.2 Sampling “C. tomentosum samples were […] frozen, freeze-dried and stored individually at –80 °C for biochemical analysis.” The organic phase referred named in section 4.3 Lipid extraction results from the mix of 2.5 mL of methanol and 1.25 mL of chloroform with the macroalgal biomass. This organic phase containing lipids is transferred to a new tube. This lipid phase can contain some impurities (e.g. polysaccharides, small peptides), then we add ultrapure water to create a two-phases system and remove no-lipid compounds. Organic phase (lower phase) includes lipids and aqueous phase includes impurities (no lipid compounds). We recover and dry the organic phase to lipid analysis.

In order to make clear this issue, the experimental section was clarified, and a new sentence has been introduced as following in Line 415 in tack-changes manuscript.

 “The upper aqueous phase containing no lipid compounds was discharged.”

Reviewer 2 Report

This paper describes oil components of algae, Codium Tomentosum measured by LC-MS with hilic column etc, and antioxidant activity.  Oil components were analyzed, and 296 chemicals were found including glycolipids sulfolipids and phospholipids. No new compound was found and no specific bioactive compound was found. But several useful informations applying for healthy nutritional diet was found. 

The oil composition of Saturated, MonoUF,n6,n3 of this algae is about 5,2.5,2,4, and this shows good source of n3 PUSFAcid. Also showed the seasonal change of these oils. Therefore, the results shows potential material of this algae for improving fat consumption habit for western style dishes. 

However, authors described in introduction, that the aim of this study is to ascertain how the chemical ecology of C. tomentosum is shaped by different growth conditions and habitats, analysing and characterizing its lipid extracts and antioxidant activity.
The authors have no point of nutritional view. It is necessary to describe the nutritional merit of Saturated, monoUA, Polyunsaturated n3 and n6 fatty acid. So it is better to rewrite specific aim of the paper for fitting to this journal.

Authors used hilic column as high resolution, but no data is available. Show one of LC data that shows high resolution as supplemental fig.

Cluster analysis etc. is interesting, but it is difficult to understand why it is necessary these analysis for specific aim of this research for nutritional points. Please describe.  

Author Response

Reviewer #2

This paper describes oil components of algae, Codium Tomentosum measured by LC-MS with hilic column etc, and antioxidant activity.  Oil components were analyzed, and 296 chemicals were found including glycolipids sulfolipids and phospholipids. No new compound was found and no specific bioactive compound was found. But several useful informations applying for healthy nutritional diet was found. 

The oil composition of Saturated, MonoUF,n6,n3 of this algae is about 5,2.5,2,4, and this shows good source of n3 PUSFAcid. Also showed the seasonal change of these oils. Therefore, the results shows potential material of this algae for improving fat consumption habit for western style dishes. 

However, authors described in introduction, that the aim of this study is to ascertain how the chemical ecology of C. tomentosum is shaped by different growth conditions and habitats, analysing and characterizing its lipid extracts and antioxidant activity.

R2C1 - The authors have no point of nutritional view. It is necessary to describe the nutritional merit of Saturated, monoUA, Polyunsaturated n3 and n6 fatty acid. So it is better to rewrite specific aim of the paper for fitting to this journal.

Authors’ reply:  We understand the point of view of the Reviewer, but the nutritional value evaluation was not the goal of our work .The nutritional value of C. tomentosum was already described in previous studies, then this was not the aim of the present study. The objective of this study was to identify the spatial and seasonal variations in polar lipidome, pigments and antioxidant activity. This aim fits better with the scope of the journal Marine Drugs.

R2C2 - Authors used hilic column as high resolution, but no data is available. Show one of LC data that shows high resolution as supplemental fig.

Authors’ reply: We have added a supplementary figure (Supplementary Figure S1) with LC chromatograms in positive and negative modes. Please take into account that due to this new figure, Supplementary Figure S1 in the original manuscript corresponds with Supplementary Figure S2 in the revised version.

R2C3 - Cluster analysis etc. is interesting, but it is difficult to understand why it is necessary these analysis for specific aim of this research for nutritional points. Please describe.  

Authors’ reply: As it was referred in the comment R2C1, evaluation of Codium tomentosum nutritional value was not the goal of the present study. To evaluate the chemical plasticity of Codium tomentosum we used  multivariate analysis, that is a widely used and accepted tool for samples discrimination. Clustering analysis and PCO allow to identify the differences between sampling groups. Additionally, heatmaps allow to identify the molecular species that more

Reviewer 3 Report

Dear Editor,

I carefully read the manuscript by Rey et al., which fits with the main interests of the Journal. The study is original, even if the manuscript contains several typos, so that it needs to be carefully revised by a native-English speaking person.

Some comments for the Authors in order to improve their paper:

  • Statistical analysis - Authors should specify how was the sample size assessed.
  • Statistical analysis - Authors should specify how was evaluated the normality distribution of the considered parameters.
  • In the discussion, Authors should specify the limitations of their study.
  • Most references are dated and should be replaced with newest article. 
  • In the figures 2, 3, 5 and 6 the values of the scales should be reported vertically, because they are now difficult to read.

Author Response

Reviewer #3

R3C1 - I carefully read the manuscript by Rey et al., which fits with the main interests of the Journal. The study is original, even if the manuscript contains several typos, so that it needs to be carefully revised by a native-English speaking person.

Authors’ reply: We have revised the manuscript.

Some comments for the Authors in order to improve their paper:

R3C2 - Statistical analysis - Authors should specify how was the sample size assessed.

Authors’ reply: We acknowledge the comment of the Reviewer. We were unable to determine the sample size before the study. To perform power calculations, it is necessary to have, at least, an idea of the variability of the measured independent variables, such as its standard deviation or, if we want to compare two proportions, an estimative of proportions in the control group. As this is the first study in this area, this data is not available.

R3C3 - Statistical analysis - Authors should specify how was evaluated the normality distribution of the considered parameters.

Authors’ reply: This information was already described in the original version of the manuscripts.

“Assumptions of normality and homogeneity of variance were verified prior to analysis through Shapiro-Wilks and Levene’ s tests, respectively.”

(Line 541 in the original manuscript version)

R3C4 - In the discussion, Authors should specify the limitations of their study.

Authors’ reply: The Reviewer #3 has not specified the limitations, we believe that in this comment the Reviewer refers that the sampling was performed only in two periods, then we have added the following sentence in the section 3. Discussion

“An annual sampling will allow to survey lipid dynamics along seasons and estimate upwelling influence in wild population lipidomes.” (Line 255 in tack- changes manuscript)

R3C5 - Most references are dated and should be replaced with newest article. 

Authors’ reply: We understand the suggestion of the reviewer, but this is still an unexplored field and references to be cited from the last years are scarce.

R3C6 - In the figures 2, 3, 5 and 6 the values of the scales should be reported vertically, because they are now difficult to read.

Authors’ reply: As suggested, the orientation of the Y axis labels has been changed.

Round 2

Reviewer 2 Report

I still feel you should describe the nutritional advance from the data of oil chemical composition.